# Sustainable Drive Tourism Routes: A Systematic Literature Review

Sandra P. Cruz [1,2,*], Cláudia Ribeiro de Almeida [3], Pedro Pintassilgo [4] and Ricardo Raimundo [5]

1    Faculdade de Economia and CEFAGE, University of the Algarve, Campus de Gambelas,
     8005-139 Faro, Portugal
2    ESCAD—Escola Superior de Ciências da Administração, IPLUSO, 1200-427 Lisboa, Portugal
3    CinTurs, Research Center for Tourism Sustainability and Well-Being, ESGHT—Superior School of
     Management, Hospitality and Tourism, University of the Algarve, Campus da Penha, 8005-139 Faro, Portugal
4    CinTurs, Research Center for Tourism Sustainability and Well-Being, Faculty of Economics,
     University of the Algarve, Campus de Gambelas, 8005-139 Faro, Portugal
5    ISEC Lisboa, Instituto Superior de Educação e Ciências, 1750-142 Lisboa, Portugal
*    Correspondence: spcruz@ualg.pt

**Abstract:** Drive tourism (DT) has become an attractive way to visit tourism destinations for an increasing number of visitors along driving routes. This flow of visitors has made sustainability a major issue, that is, the way by which tourism development ensure economic benefits for local communities and preserves local identity, along the route, without compromising the environmental resources. Many studies focused the topic of DT, mainly the analysis of a particular angle, either be economic sustainability, e.g., advantages of the ones related to economic and environment sustainability, such as the impact of tourists along the route environment. Nevertheless, little attention has been paid to the social consequences of DT in the local entrepreneurial environment and the resulting exaggeration of their cultural representativeness in the sense of authenticity. Our aim is to summon these points of view and achieve, through a systematic literature review, a clear and integrative picture of the driving tourism impacts in terms of sustainability along the routes throughout local communities. A systematic literature review was performed using the PRISMA guidelines (Preferred Reporting Items for Systematic Reviews and Meta-Analyses) methodology. This systematic literature review sought to consolidate knowledge on the subject. In order to illustrate the link between major categories and their corresponding trends, authors used *VOSviewer* scientific software. The gathering of existing knowledge around the three components of sustainability highlighted the importance of community involvement and collaboration among DT stakeholders to address the trade-off between the protection and promotion of DT routes. Opportunities for future studies are suggested.

**Keywords:** drive tourism; routes; sustainable; systematic literature review; PRISMA

## 1. Introduction

The environment, social justice, and development have become an important issue worldwide and one of the main topics of analysis. A change in values has been carried out in order to ensure the sustainability of future generations, while triggering change with regard to behaviors and raising consciousness about sustainability issues.

All sectors, including tourism, face several challenges in order to achieve sustainable goals, with investments on infrastructure, processes, procedures, and equipment that support this change (Jiang and Lyu 2022; Ooi et al. 2018; Hanrahan et al. 2017; Guizzardi et al. 2022). Tourism agents struggle to balance the environmental, economic, and social priorities that are continuously changing and evolving.

Factors affecting climate change, economic instability, and other macro-environmental issues have profound implications on economic ecosystems and to society with unknown impacts for the future.

To balance this, tourism stakeholders face new challenges on a daily basis, connected not only to the new trends that emerged in the last decade, but also with the need to create new strategies and business solutions that balance environmental, economic, and social issues. Therefore, tourism is also changing and evolving rapidly in both developed and developing regions (Ooi et al. 2018).

Sustainable tourism comprehends optimizing the exploitation of environmental resources to preserve natural, cultural heritage and ensuring authenticity of the host community, while ensuring socio-economic benefits in the long term. Therefore, it requires sustainable planning to maximize community benefits and minimize community costs thus increasing community participation (Hanrahan et al. 2017). According to UNWTO (World Tourism Organization), sustainable tourism could be defined as "takes full account of its current and future economic, social and environmental impacts, addressing the needs of visitors, the industry, the environment and host communities". Its objectives encompass improving the quality of life of the host population both in the short and long term, fulfilling the tourists' demands, and protecting nature. Literature posits that a superior conservation of the heritage is the central indicator of perceived sustainability, whereas tourism could be a trigger for destinations' competitiveness in terms of sustainability (Guizzardi et al. 2022). However, there a few studies published in sustainable drive tourism.

Conflicting perspectives of sustainable tourism subsist (Buffa 2015; Filimonau et al. 2022) Some researchers argue that sustainability is not constantly compatible with tourism because there are too many competing interests that end up with certain interests overriding others (Filimonau et al. 2022). Moreover, new small businesses have thrived based on local cultural and natural heritage, while feeding local entrepreneurship (Filimonau et al. 2022). Some literature claims that sustainable tourism can become a marketing label assumed by destinations to draw an increasing number of visitors who are aware about sustainability concerns (Guizzardi et al. 2022; Buffa 2015).

Several authors (Hanrahan et al. 2017; Buffa 2015; George et al. 2013; Taylor and Carson 2010; Fjelstul and Fyall 2015) mention that sustainability is a strategic goal that needs to be achieved by all tourism destinations, besides their scale of geographical area. To accomplish this goal, destinations need to amend the objectives of tourism, as a facilitator of cross-cultural commitment, ecological enjoyment, and spiritual development rather than selfish and hedonistic modes of tourism (Fjelstul and Fyall 2015; Saluja et al. 2022). Sustainability is one of the main competitive factors for tourism destinations, creating value for tourists and to the overall community (Laws and Scott 2003; Saluja et al. 2022). In addition, sustainable tourism becomes a comparative advantage that prompts economic growth, within a bidirectional relationship between the two variables (Brida et al. 2015).

Tourism destinations face constant challenges, and measures related to sustainable tourism must be adapted depending on demand, supply, and mainly host communities' needs (Butler et al. 2021a; Sykes and Kelly 2016). A central condition is a cultural change inherent to values and behaviors that may trigger new visitors, businesses, and associated organizations that can lead in turn to positive behaviors to enhance safe and enjoyable experiences (Buffa 2015; Butler et al. 2021a; Wu 2015). Sustainability should be shared in the tourism actors' goals towards sustainable behaviors such as the case of motorcycle tourism, in which dynamic interaction leads to a new motorcycle leisure lifestyle whilst ensuring new business demand (Sykes and Kelly 2014).

Additionally, tourists should be conscious of sustainability issues, whilst hosting communities should aim to ensure cultural and natural integrity, reinforce the tourists' connection to the destination through memorable experiences, and allow the reinforcement of identity creation (Marschall 2012).

Sustainable tourism is particularly interesting when examined in sites of DT (Taylor and Carson 2010; Cartan and Carson 2009; Taylor and Young 2005). The term DT is used here to refer to tourism routes connecting the city to the rural areas by linking a variety of activities and attractions, that in turn stimulate entrepreneurial opportunities through the development of ancillary products and services along the route, integrated to support

the development of a region, conservation, and rehabilitation of cultural and natural resources (Zakariya et al. 2020). It comprehends special tourists whose travel motivation is to experience the rural environment, e.g., wine routes or storehouses routes and entails the desire for personal escape, experiencing the flow state while riding down the routes, for example, in the case of motorcycle tourists (Frash et al. 2018).

DT routes exhibit an inherent cultural (e.g., archaeological artifacts) and recreational value (e.g., exploring landscape), in which tourist satisfaction positively leads to the intention to revisit sites (Qiu et al. 2018) and connect regions (Shih 2006). DT combines elements of diverse tourism trends (e.g., visits to industrial and natural sites), as almost a quarter of all recreational trips (over 60 miles) are taken in private vehicles. Despite this fact, there is a scarcity of studies in the tourism literature covering the topic of DT (Shanahan 2003).

The flow of visitors entering DT routes leads to a need for balancing in terms of sustainability, for example, in how to ensure a trade-off between conservation of landscape and tourism promotion (Prideaux et al. 2001; Dou et al. 2022). Despite the significance of this issue, several studies do not explicitly mention sustainability goals, favoring the economic development instead. In this regard, quite a few studies have so far focused on environmental impact, in terms of conservation of natural resources (Filimonau et al. 2022; Larsen et al. 2020; Wu et al. 2018) by discussing factors such as the impact of air quality on visitors (Wu et al. 2018). Other authors investigated economic implications (Rolfe and Flint 2018), for example, about the potentialities of rural contexts (Ramsey and Malcolm 2018). The social implications have been disregarded in such studies though (Ramsey and Malcolm 2018), while ignoring that without the commitment of locals, the goal of sustainable tourism may not be achieved (Fjelstul and Fyall 2015). The aim of this paper is to summon these diverse perspectives and attain, through a literature review, a clear picture of the sustainability of tourism along the DT routes.

In the following sections, we first describe the method adopted for the review of existing studies that have analyzed the sustainability tourism issues in DT routes. Next, based on a deductive coding process, we discuss existing knowledge according to the three components of sustainability, that is, economic, environmental, and social dimensions through three different sections. The last section concludes and delivers conclusive insights.

## 2. Materials and Methods

The research method adopted in this study is the systematic literature review, allowing the definition of a research boundary that should be developed from a scientific perspective, constituting a transparent, replicable, and scientific process, which aims to minimize biases through exhaustive bibliographic research publications and unpublished studies. Thus, after the formulation of research questions, the PRISMA (Preferred Reporting Items for Systematic Reviews and Meta-Analyzes) methodology was applied. The references papers selection process consists of four stages: (1) databases selection, (2) papers extraction, (3) abstract screening and (4) full-text screening. The selection process is explained using the PRISMA 2020 flow diagram (Figure 1) (Moher et al. 2009). The research formulated three research questions related to drive tourism, the tourist segments, the relationship between sustainability and drive tourism routes, and the emergent entrepreneurship and subsequent quality of life. The databases used for consultation were Scopus by Elsevier and Web of Science (WoS) by Clarivate.

The research method was developed starting with the keyword that appears in abstracts "Drive Tourism" with the Boolean term "OR" with keyword "Self-drive tourism", since it was noted that the researchers do refer this keyword with the same meaning to drive tourism. We add another keyword ("routes entrepreneurs") with the Boolean term "OR", since we are interested in perceiving the small business along the route. Finally, we limited our research only to scientific articles with peer review (LIMIT-TO DOCTYPE, "ar") to guarantee that we are searching without differences. Overall, 99 potentially selectable contributions were identified within the database of "Scopus" and 45 contributions were found within the database "web of Science". Only scientific papers written in the English

language and published in business, tourism, heritage, economy, hospitality, environment, and management areas were selected (Figure 1).

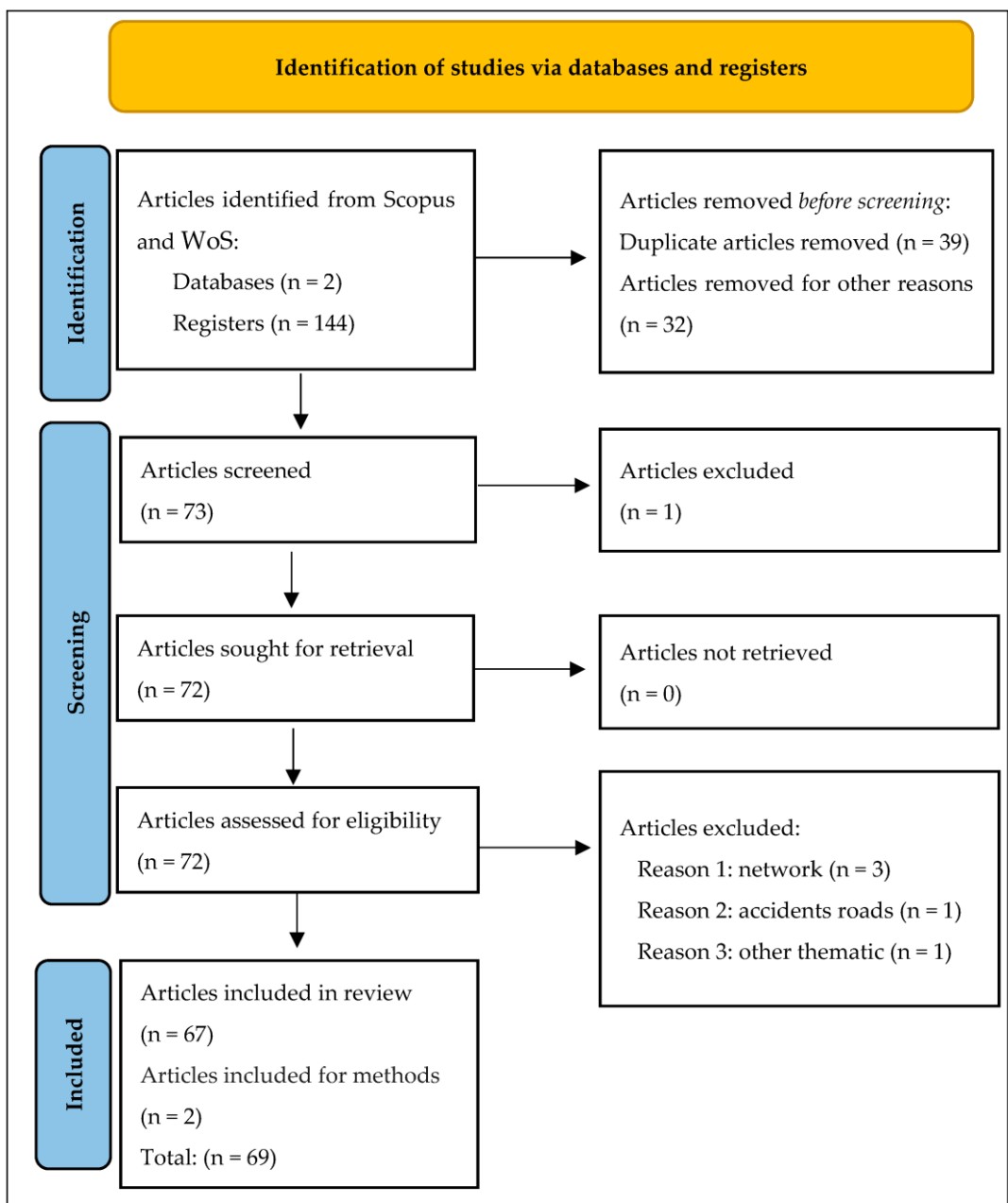

**Figure 1.** PRISMA 2020 flow diagram (source: own elaboration).

In order to illustrate the link between major categories and their corresponding trends, authors used *VOSviewer* scientific software. *VOSviewer* is a software tool for creating maps based on network data and for visualizing and exploring these maps (Waltman et al. 2010). Items may be grouped into clusters. A cluster is a set of items included in a map. Clusters are non-overlapping in *VOSviewer*. In other words, an item may belong to only one cluster. Clusters do not need to exhaustively cover all items in a map (Waltman et al. 2010) Hence, there may be items that do not belong to any cluster.

## 3. Results

*Literature Analysis: Themes and Trends*

Authors analyzed peer-reviewed documents on the topic in the period between January 2001 and July 2022. This analysis allowed us to understand that 2022 was the year with the highest number of peer-reviewed documents on the subject, with 14 contributions. Since 2011, the interest in research on drive tourism has had an increasing trend (Figure 2). Despite the pandemic outbreak of COVID-19 in 2020, we can observe an increasing trend.

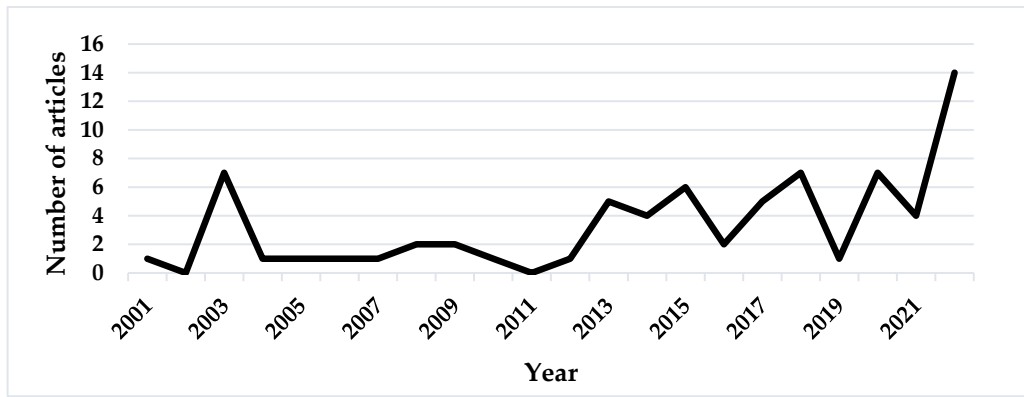

**Figure 2.** Number of articles by year. Source: own elaboration.

When researched by authors, it is possible to observe that 151 authors wrote about the topic of drive tourism, but only 15 authors published two or more articles on the subject, as for example D. Carson and B. Prideaux that published four articles about the topic. (Figure 3).

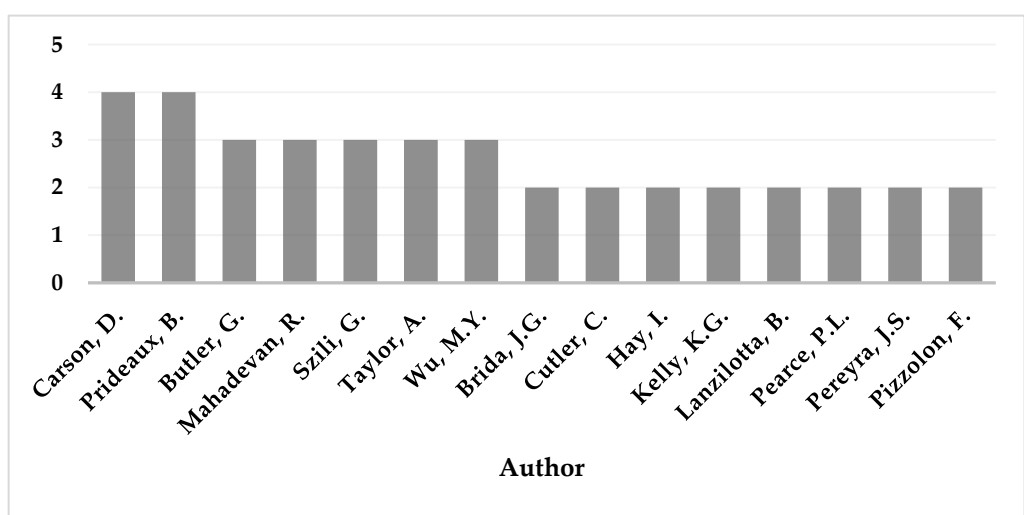

**Figure 3.** Authors with two or more publications. Source: own elaboration.

Among all analyzed journals, the Journal of Vacation Marketing with nine articles has the highest number of published articles on drive tourism, followed by Tourism Analysis, Journal of Destination Marketing and Management, Tourism and Journal of Heritage Tourism, Tourism Recreation Research and Proceedings of N. Academy of Sciences (Figure 4).

When analyzed by country, it is possible to see that Australia, USA, China, South Africa, United Kingdom, and Italy were the top for articles about "Drive Tourism" (Figure 5).

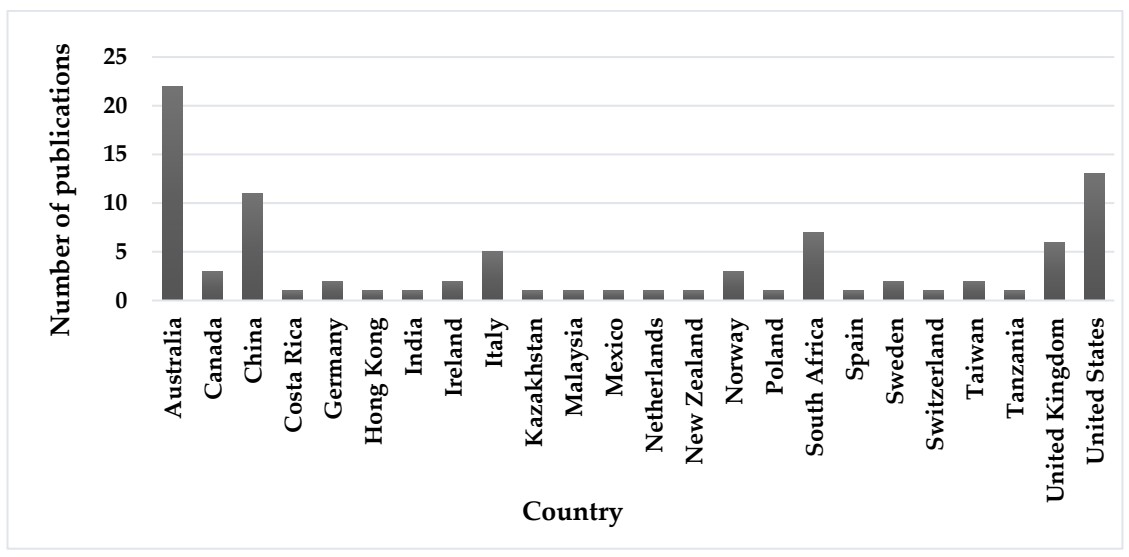

**Figure 4.** Number of publications by journals Source: own elaboration.

**Figure 5.** Number of publications by country. Source: own elaboration.

In Table 1, we represent the Scimago Journal and Country Rank (SJR), the best quartile, and the H index by publication. "Proceedings of The National Academy of Sciences of The United States of America" was the highest, with 805 (SJR), Q1, and the H index was 4.18.

**Table 1.** Scimago journal and country rank impact factor. Source: own elaboration.

| Title | SJR 2021 | Best Quartile | H Index |
|---|---|---|---|
| Journal of Vacation Marketing | 0.6 | Q1 | 68 |
| Journal of Destination Marketing and Management | 1.75 | Q1 | 50 |
| Journal of Heritage Tourism | 0.82 | Q1 | 36 |
| Tourism Analysis | 0.65 | Q2 | 39 |
| Tourism Recreation Research | 0.88 | Q1 | 50 |
| Asia Pacific Journal of Tourism Research | 0.89 | Q1 | 44 |
| Journal of Sustainable Tourism | 2.48 | Q1 | 114 |
| Journal of Tourism and Cultural Change | 0.63 | Q1 | 31 |
| Journal of Travel Research | 3.29 | Q1 | 145 |
| Tourism Geographies | 2.27 | Q1 | 73 |
| Tourism Management | 3.38 | Q1 | 216 |
| African Journal of Hospitality Tourism and Leisure | 0.21 | Q3 | 14 |
| Australian Geographer | 0.83 | Q1 | 48 |
| Biological Conservation | 2.14 | Q1 | 213 |
| Canadian Geographer | 0.47 | Q2 | 48 |
| Current Issues in Tourism | 1.84 | Q1 | 82 |
| Economic Analysis and Policy | 0.77 | Q1 | 34 |
| Economic Modelling | 1.07 | Q1 | 87 |
| Food Quality and Preference | 1.15 | Q1 | 129 |
| International Journal of Culture Tourism and Hospitality Research | 0.6 | Q2 | 36 |
| International Journal of Geoheritage And Parks | 0.37 | Q2 | 7 |
| International Journal of Hospitality and Tourism Administration | 0.62 | Q2 | 36 |
| International Journal of Tourism Research | 1.14 | Q1 | 67 |
| Journal of Mountain Science | 0.55 | Q2 | 41 |
| Journal of Organizational Computing and Electronic Commerce | 0.65 | Q2 | 43 |
| Journal of Policy Research in Tourism Leisure and Events | 0.7 | Q1 | 27 |
| Journal of Sport and Tourism | 0.53 | Q2 | 46 |
| Journal of Tourism History | 0.26 | Q1 | 11 |
| Journal of Travel and Tourism Marketing | 2.05 | Q1 | 82 |
| Journal of Travel Research | 3.49 | Q1 | 145 |
| Land Use Policy | 1.64 | Q1 | 125 |
| Leisure Studies | 0.67 | Q1 | 69 |
| Online Information Review | 0.63 | Q1 | 64 |

**Table 1.** *Cont.*

| Title | SJR 2021 | Best Quartile | H Index |
|---|---|---|---|
| Planning Malaysia | 0.26 | Q2 | 9 |
| Proceedings of The National Academy of Sciences of the United States of America | 4.18 | Q1 | 805 |
| Rangeland Journal | 0.45 | Q2 | 42 |
| Revista de Economia Mundial | 0.2 | Q1 | 13 |
| Scandinavian Journal of Hospitality and Tourism | 1.17 | Q1 | 50 |
| Sustainability Switzerland | * | * | * |
| Tourism Culture and Communication | 0.32 | Q1 | 16 |
| Tourism Economics | 1.04 | Q1 | 64 |
| Tourism Planning and Development | 0.84 | Q1 | 36 |
| Tourism Review | 148 | Q1 | 38 |
| Tourismos | 0 | (a) | 20 |
| Tourist Studies | 0.93 | Q1 | 50 |
| Transportation Research Part F Traffic Psychology and Behaviour | 1.46 | Q1 | 100 |
| Tropical Geography | 0.19 | Q3 | 5 |

**Note**: * data not available; (**a**) Not yet assigned quartile. **Source**: own elaboration.

There was a total of 54 contributions in Q1, 9 contributions in Q2, and 2 contributions without quartile. There was a total of two contributions in Q3 and no publications in Q4. Articles retrieved with quartile Q1 represent 80.6% of the total publication and for best quartile Q2 represents 13.4% of the total publication.

The bibliometric study is displayed to investigate and identify indicators on the dynamics and evolution of scientific information. The study of bibliometric results, using the scientific software *VOSviewer*, aims to identify the main research keywords in studies focused on drive tourism (Figure 6). Keywords provide by authors of the paper and occurred more than four times in the WOS core database were enrolled in analysis. *VOSviewer* provides eight clusters, referred to as the network visualization of bibliometric study. The keywords that appeared most were "DT", "rural tourism", "road", "automobility", "mobility", "Conservation", "Heritage management", "accomodation", "travel", to refer to the principal terms. The size of nodes indicates the frequency of occurrence. The curves between the nodes represents their co-ocuurence in the same publication. On the other hand, the shorter the distance between the nodes, the larger the number of co-occurrence of the two keywords (Figures 6 and 7).

*VOSviewer* provides the visualizations referred to as the overlay visualization. The author keywords can be examined in Figure 7, making clear the network of keywords that appear in each scientific article, thus allowing us to know the topics studied by the researchers and identify future research trends. Thus, we can observe that the "Drive tourism" keyword was published linked to keywords such as opportunity, pandemic, economic development, country, community engagement, motorcycle backpacker, to refer to the principal terms.

The network visualization for citation authors is in Figure 8. We can highlight several authors, writing on drive tourism from different but connected perspectives. This bibliometric analysis of citations shows eight clusters in different colours for the most cited author.

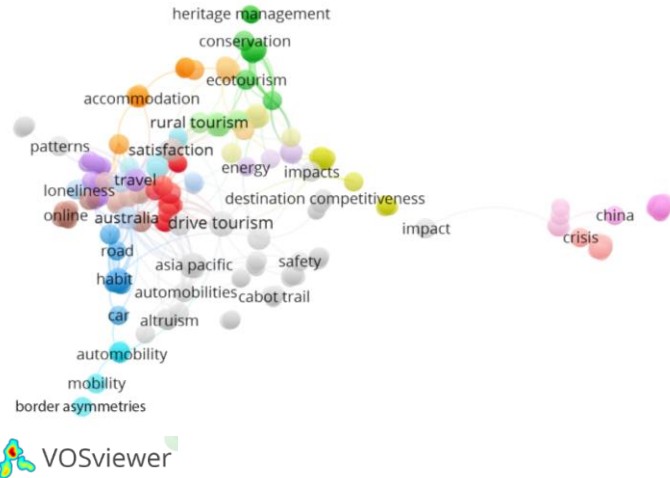

**Figure 6.** Network visualization for all keywords.

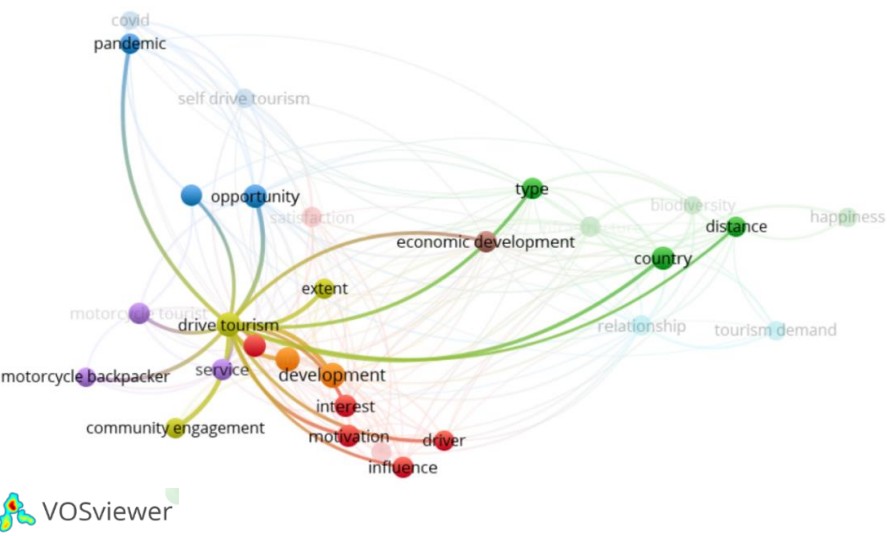

**Figure 7.** Overlay visualization for the Keyword "Drive tourism".

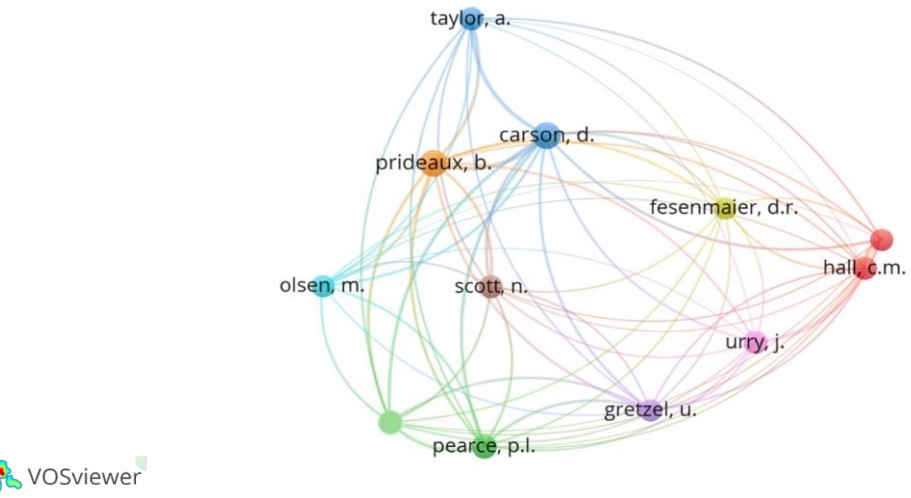

**Figure 8.** Authors Networks of bibliographic citations.

## 4. Discussion

In the following sections, we have gathered insights from pieces of literature included in this review and set the analysis around the three dimensions of sustainability: economic, environmental, and social (Guizzardi et al. 2022).

### 4.1. Economic Dimension

Some authors focus on the economic characteristics of sustainability, pointing out the importance of DT as an added tourism attraction to the destination, with positive impacts on the local economy, creating job opportunities, encouraging investment in new businesses, in particular in rural areas, while maintaining the destination attraction through a collaborative management effort (Lemky 2017). It is notable the existence of an interplay between real per capita GDP and tourism (Lemky 2017), as tourism activity leads—in the long term—to economic growth, or, on the other hand, economic development drives tourism growth, being apparent a bidirectional interplay (Brida et al. 2015). For example, scenic travel routes created to provide opportunities for tourism and recreation and to encourage economic development, in particular in rural areas, while maintaining the destination attraction through a collaborative management effort (Lemky 2017). Despite the positive economic implications, it will avoid several environmental issues (Lemky 2017).

A strategic planning is needed with regard to DT routes. Authors argue that, depending on destinations and geographical areas, the public road administration and route planning procedures are different, using Norway as an example that implemented a top-down principle regarding the labeling of routes. On the contrary, in Sweden, the standard is of muddling through, giving street-level planners more opportunities for individual influence on both the route and the surrounding area planning (Antonson and Jacobsen 2014), to better supply the DT market (Sivijs 2003).

Some studies about the economic dimension of sustainability are more connected to the management dynamics of the supply side, while analyzing the pertinent characteristics of visitors, to better ascertain the impact of tourism on the local community and ensuring alternative strategies of livelihood. A good example is India's first National Geopark in Varkala cliffs, a well-established tourism destination, for both domestic and international travelers, that constitutes the major source of livelihood for the local communities, despite the issues related to seasonality (Saluja et al. 2022). In this vein, dinosaur fossils provide a potential resource for remote-region economic development through commoditization as a new tourism attraction and new tourism services (Laws and Scott 2003). With respect to supply side, network strategies and marketing policies are suitable to promote the attractiveness of these DT routes and thus generate economic benefits in the surrounding areas, as in the case of the economic benefits of an access road to encourage tourism at deserts or at coast (Rolfe and Flint 2018). However, the introduction of a new product such as *4Whel Drive market* as a new economic strategy, developed in Australia, does not always have positive net economic benefits for the local community (Cartan and Carson 2009).

Regarding the factors that influence the attractiveness of DT routes, authors suggest, for example, the proximity to other tourism attractions and tourism segments (Buffa 2015; Laws and Scott 2003; Fjelstul and Fyall 2015), physical infrastructures, location, access, attractions, promotion, accommodation, and the history of the place (Saluja et al. 2022; Butler et al. 2021a; Marschall 2012).

Other authors mentioned that national cooperation and coordination is mandatory, and helps contribute to a territorial image, which is important to point out authenticity and sustainability characteristics and to create common strategies to attract and retain the visitors in the route for more time (Qiu et al. 2018; Antonson and Jacobsen 2014).

Regarding the transport infrastructures, it is important to evaluate the physical conditions of highway and tolls price, as these can be determinant factors for the tourist to feel more attracted to travel by other modes of transport, for example, high-speed railway. Literature also suggests that factors that influence the attractiveness of DT routes include proximity to other tourism attractions and tourist segments.

For many tourists, the DT routes themselves are not the main motivation for travel, as normally tourists add some other visits and territorial attractions to the main experience. For this reason, it is very important to create common strategies between private and public entities, to promote and develop a solid product that values the main characteristics of the route and the territory, their attractions, services, history, and other elements that may be important for tourist decision. In this framework a route manager could improve the potential of the DT routes and develop tools to gather data concerning the visitor profile, expenditures, economic impacts, and others that bring new information and knowledge (Prideaux and Carson 2003).

National cooperation and coordination are paramount, as visitor road corridors have shown to be central to visitor dispersion, while relying on cross-sector tourism cooperation, as in the case of themed routes, deemed as 'win-win' tourism outcomes, thus highlighting the need for greater national coordination (Olsen 2003). In the same vein, one can weigh against the impact of two different transport infrastructures, highway and high-speed railway, on tourist flows, in which tourism via high-speed railway was responsive to the position of trip destination, whilst self-drive tourism was more susceptible to travelling time (Liu et al. 2022). Noteworthy, this strategy, combined, contributes to the creation of a "territorial image" that emphasizes the importance of authenticity and sustainability (Olsen 2003).

Some authors state that local cultural traditions and previous experiences underpin diverse types of capital and shape entrepreneurship in decisive times, as was the case for women tourism entrepreneurs during COVID-19 (Filimonau et al. 2022). For example, Morden, a small city in Manitoba, Canada, has been lately doing well in diversifying its economy, including hospitality, manufacturing, services, and tourism. This was partly due to a south-central Manitoba location and to an innovative local entrepreneurship attitude towards tourism (Ramsey and Malcolm 2018).

Regarding the demand size of DT routes, studies found different typologies of tourists with 4WD (four-wheel-drive)—for example, the explorer-traveler that feels more attracted to desert areas, or the independent travelers, or multiple-vehicle-trips travelers, or also tag-along-tours travelers (Taylor and Prideaux 2008). Moreover, it could also include active/adventure tourism and geo-tourism; among other business initiatives, such as motels that can improve the mix of attributes, they are advertising to attract drive tourists along the route (Shanahan 2003). As a consequence of distinct tourist segmentation, literature suggests that the number of activities in which drive tourists participate depends significantly on the segment they belong to (Pennington-Gray 2003).

DT routes demonstrate a great tourism potential because of the heterogeneous interest amongst driving tourists. Its strength relies mostly on the aforementioned development of attractions, a profound understanding of the drive tourist, community involvement, effective interpretation, and infrastructure (Hardy 2003). The output could be substantial economic return, for example, through an increasingly entrepreneurial attitude within communities (Zheng et al. 2016). Driving tourists can contribute to the improvement of the local economy by staying in accommodations along the road, visiting local villages, and buying local products related to the local heritage, in which senior tourists constitute a central segment (Prideaux et al. 2001).

The driving forces behind tourists' travel choices, the main motivational influences include the destination attractiveness, the desire to enhance one's relationship and socialization, discover new places, and experiencing feelings of enjoyment (Buffa 2015; Wu et al. 2018; Patterson et al. 2015). Motorcycle tourists are a good example of customers that value infrastructures, environment, hospitality, and good services (Frash et al. 2018) in DT routes. They are good customers, who normally return to the same routes and bring others with them. DT routes managers need to look for the different tourist profiles and develop strategies and promotional campaigns accordingly, in order to capture their attention and visit overtime, with direct impacts on the local economy of the area (Frash et al. 2018). In so, there is a need to develop segmentation strategies that match the types of tourists

targeted by destination (Tkaczynski and Rundle-Thiele 2019), for both international and domestic tourism (Lin et al. 2020; Leick et al. 2021; Tripathi and Shaheer 2022).

*4.2. Environmental Dimension*

Environmental issues are also very important to evaluate sustainability. Several authors (Ooi et al. 2018; Taylor and Carson 2010; Saluja et al. 2022; Dou et al. 2022; Echeverri et al. 2022) argue that the development of sustainable tourism should be based on the suitable usage of natural resources and the cautious improvement of natural processes of the sites. In terms of DT routes, several authors mention the importance of the environmental issues that need to be balanced in order to protect natural resources and assure that new investments can provide both biodiversity conservation and positive economic impacts for the local community.

There are environmental goals that guide the development of tourism, aimed at enhancing and protecting the environment of DT routes. There is a need to balance tourism with the protection of the natural resources. Yet, the degree to which biodiversity goals drive tourism, especially with respect to infrastructure, is poorly understood, while investments in infrastructure must keep up with successful biodiversity conservation for tourism to create attractive economic revenue (Echeverri et al. 2022).

Regarding the relationship between tourism development and the environment, although the latter is sometimes identified as a restraint, it turns out that it often enhances destinations' competitiveness instead (Guizzardi et al. 2022). For example, the migration of more than a million wildebeest in the Serengeti-Mara, in Africa ecosystems, generates economic benefits through ecotourism and strengthens the continued conservation of ecosystems that contain wildlife resources (Larsen et al. 2020).

It is worthwhile to identify and categorize all the elements present in DT routes that can attain an important impact at the environmental level, and are representative of the site, of scientific and recreational interest. It is valuable to acquaint the scenery, road facilities, and available activities that might have a significant impact on drivers' satisfaction. For example, for the Chinese drive tourists, the responses of the local community to their trip, as well as central environmental issues, in particular air quality, are peculiarly key concerns (Wu et al. 2018).

There are ways that accomplish a balance between tourism enhancement and the protection of natural resources, such as the cooperation between local actors in order to develop a sustainable model of tourism, that protect the main environmental characteristics of the areas along the route and contribute in a positive way for the overall community. The tourism industry can collectively respond and adapt to changes, based on human interactions with sensitive ecosystems through resiliency, innovation, and adaptation, allowing us to combine natural issues of the route and their cultural value. These measures can improve drive tourists' experience, thus allowing for tourism development (Ooi et al. 2018; Fjelstul and Fyall 2015).

Cooperation amongst the diverse stakeholders (scientists, local authorities, owners—public and private institutions) is needed to distinguish the potential of the natural resources and improve the safety of the environment—for example, for the people who drive through natural environments, often at fast speeds and more destination-oriented, whose interest for the sites along the route is relative and likely not fully exploited (Ooi et al. 2018; Saluja et al. 2022)

Another way of interest to lead a balanced support of the natural resources in DT routes relies on the tourists' profiles in terms of educational and demographic segmentation that impact on their decision-making processes, motivations, and behaviors. For example, a distinction is made between hard path young tourists and soft path young tourists. Their different profiles should be deemed in destination strategies, as the strong sympathy of the former to sustainability suggests the likelihood of developing offers that optimize some distinctive features of a territory (Buffa 2015). Thus, planning the DT routes is demanded to open further paths, able to include the needs of several actors, such as decision makers,

residents, local firms, and tourists for the management and preservation of DT routes (Fjelstul and Fyall 2015).

### 4.3. Social Dimension

The social dimension of sustainability comprehends a social viewpoint to approach the socio-cultural outcome of tourism development. Consumers are deemed as identity seekers, in which the sensory experience of tourism creates a unique link for visitors with the destination, therefore providing memorable and, thus, authentic experiences (Esau and Senese 2022). In this way, sustainability adds value to the input of people to tourism development and to the development of the DT routes to accomplish the growth of the local economy and ensure the approval of tourists' demand. Sustainability also embraces the impacts of tourism development on ameliorating the quality of life of the local communities in the long range, emphasizing their community identity and authenticity, whilst linking tourists' happiness with the local quality of life at a destination (Jiang and Lyu 2022; Paniagua et al. 2022).

A literature review allows us to understand that social cultural repercussions in DT routes is connected to the creation of community identity and collective participation in the decision-making process of tourism development. Some authors refer the potential of DT routes to the economic revitalization of less attractive regions (Wu 2015; Liu et al. 2022; Taylor and Prideaux 2008), mainly because these territories are identified and shared by different visitors, on social media or even on live streams (Saluja et al. 2022). Visitors play and important role in the informal promotion of a DT route, improving the knowledge of the areas for others that are not so familiar with them.

The potential of the DT routes with respect to the social and economic revitalization of previously tourism, less attractive regions has been mentioned by studies of this topic (Jiang and Lyu 2022; Paniagua et al. 2022; Li et al. 2022). Some territories were put on the map by tourism live streamers' while sharing their travel experience, in terms of entertainment and self-presentation, in which monetary incentives are identified as a central motivation. (Li et al. 2022). Additionally, by improving eco-tourism practices throughout the route, mainly in peripheral regions less developed (Ramsey and Malcolm 2018), allow the development of some adventure tourism activities, being more attractive and allowing the improvement of roads and creating some synergies along the route between destinations (i.e., cities, villages) (Qiu et al. 2018). Either the escape to an attractive destination, or the appeal of the rally itself, the desire to socialize, was leading motivational influences (Wu and Pearce 2017).

Authors argue that tourism development can improve the quality of life of hosting communities, suggesting that drive tourism could create a community engagement and garner their support, mainly if they perceive that public entities and route managers to be creating strategies under sustainable principles with correct planning measures that could benefit the overall community (Fjelstul and Fyall 2015; Carson and Taylor 2008; Dou et al. 2022). Furthermore, scenic travel routes have been developed to offer opportunities for tourism and to promote the economic development of rural areas. However, maintaining the site attraction requires a collaborative destination management effort (Lemky 2017).

With the involvement of all the main actors and a collaborative destination management effort, a DT route can allow for the improvement of job opportunities and development of rural areas and local and familiar businesses. Some authors believe that local entrepreneurship can allow the development of a more sustainable quality of life of the residents (Filimonau et al. 2022; Saluja et al. 2022; Sykes and Kelly 2014; Mahadevan 2014; Armbrecht 2014). Tourism development along DT routes impacts substantially the sense of community along the host destinations, while offering visitors core cultural experiences (Patterson et al. 2015), shaped by a closer interaction with residents and their cultural traditions, enhancing their proud as a community. Also it is important to note that being accessible by car is a motive of proud for many communities, which means that they are

connected to other regions and they can be more attractive for visitors and tourists. A good example is Africa, that in the first decade of the twentieth century, through motor touring and by printing road reports became more known (Pirie 2013). Nevertheless, support to local initiative and infrastructures are sometimes scarce as in the case of parks. In particular, park capacity, to support the drive-tourism experience, in terms of caravanning and accommodation facilities (Caldicott and Scherrer 2013a). Community identity has a symbolic nature with the function of representing reality, as the constructs have been found to manage sport tourists' safety risk perceptions, in how the interrelationships amongst these constructs can positively influence repeat visitation (George et al. 2013).

Local communities believe in the significance of including DT in their identity to preserve their history, through memory, as memory is a crucial factor in choosing a destination due to its impact on the tourist experience at the destination and on the sharing of the experience with others after the trip, which contributes to the process of identity formation (Marschall 2012). Nonetheless, when these sites suggest negative memories, it is therefore realized as negative heritage, becoming crucial to create a new narrative (Marschall 2012). Moreover, it is important to develop products and experiences that reflect the motivations and experiential aspirations of their target as in the case of 4WD tourism in Australian desert areas (Taylor and Prideaux 2008), as there appear to be market segments based on motivations, activities, and demographics, which resemble a diversified marketplace (Taylor and Carson 2010). This process of integrating DT routes into the community's identity is often hampered by the difficulty in assuming DT routes as part of local identity, even though the inclusion of one or more professional rally sports teams, for example, among a community, with limited extent in terms of self-drive sports impacting and representing marketing opportunities for the host communities (Taylor and Young 2005), in the field of tourism behavior (Woodside et al. 2004).

Finally, the segment of senior travelers go on holiday, travel by car, and prefer the non-school-holiday periods for travel (Prideaux et al. 2001); whereas, younger travelers would rather seek fast driving irrespective of the time of year and aim to achieve an 'authentic' driving experience (Gross 2020). Either way, driving tourists engage in self-drive tourism due to the feelings of safety, adventure, and discovery that it offers compared with other modes of transport, (Butler et al. 2021a, 2021b), through which, after the pandemic of COVID-19, has become a tourism mode that enable tourists to travel freely (Gross 2020), by using at its best new vehicles technologies (Brida et al. 2013), including in caravans (Caldicott and Scherrer 2013b), according with each ones' economic conditions (Grechi et al. 2017), driving contexts (Thompson and Sabik 2018), and extant key factors for the successful development of touring routes (Dahl and Dalbakk 2015; Prideaux 2020).

## 5. Conclusions

Sustainable tourism is increasingly seen as an important element for tourist destinations, and DT routes can contribute to its development. For this, DT routes managers must integrate into their planning the three components of sustainability (economic, environmental, and social), emphasizing its function as a catalyst for intercultural engagement.

DT routes allow us to enhance natural and cultural resources, bringing to light cultural traditions, history, and some storytelling of the local communities. Most of the DT routes cross rural areas with sensitive environments that need to be evaluated and analyzed in deep in order to create sustainable and planned offers integrated in the territory in a positive way.

The involvement of public and private stakeholders as well as other community members is very important and urgent, as it allows for the creation of integrated measures and a sustainable development of the area and the route.

Many sites along DT routes are also considered UNESCO sites (e.g., along US Route 66), which are, of course, of interest to preserve in a sustainable way. The sustainability goal is therefore intrinsic along different sites along the routes, through which one can show up by economic, environmental, and social modes.

This review has provided a snapshot of the sustainable improvement of tourism in DT routes, related with the three dimensions of sustainability: economic, environmental, and social. This approach intends to deliver an integrative standpoint of those three perspectives.

The literature review carried out here demonstrates the pieces of economic literature related to DT routes and sustainability, emphasizing its strategic weight and opportunities in terms of investment in infrastructure and small businesses. Therefore, DT routes might create crucial economic benefits to their sites. Furthermore, there are plenty of tourism modes associated with DT routes, such as the aforementioned adventure tourism and cultural tourism.

The reviewed pieces of literature are also centered on environmental sustainability, underscoring that DT tourism growth must be based on the efficient use of natural resources, that need to be integrated with the local communities, that need to be involved in the overall process. Finally, the articles that highlight social sustainability regarding DT routes, suggest that the sites along the road carry out a strategic function in terms of regional entrepreneurship. In short, tourism improvement might stimulate local entrepreneurship, with ensuing enhancement of quality of life, while augmenting the sense of community.

To summarize, this literature review unveils some critical requirements with respect to DT routes, worth mentioning the need to prompt a collective decision-making process when it comes to DT routes development, with the involvement of all local communities and tourism stakeholders, focusing on promoting the destinations in terms of strategic policies.

This review underscores the need for an inclusive perspective to approach the DT routes issue that deems it as key to augment the attractiveness of a broader region, while considering the inclusion of all stakeholders aiming at sustainability. Furthermore, this review might be a trigger to start up a potential new decision-making outline that is able to collectively promote new sites down the route.

With regard to further research avenues, this piece of literature has shown that current studies have mostly focused on the environmental issues of DT routes. It would be worthy to study diverse sustainable modes of tourism in DT routes from varying more or less developed contexts and/or through distinct lens. Moreover, it would be worthwhile to extend this review through empirical research, as well as in diverse regional contexts such as the peripheral context of Portugal and its iconic EN2. Modeling and foreseeing the demand that may occur at EN2 route is crucial in determining how much money should be invested in developing infrastructures and business. Additionally, studies using a combination of spatial autocorrelation and georeferentiation systems or detectors could recognize patterns of drive tourism demand and provide information to create sustainable business along the routes, especially Portugal EN2 route.

**Author Contributions:** Conceptualization: S.P.C., C.R.d.A. and P.P.; methodology: S.P.C., C.R.d.A., P.P. and R.R.; validation: C.R.d.A.; formal Analysis, S.P.C.; investigation: S.P.C.; resources: S.P.C., R.R. and C.R.d.A.; data curation: S.P.C.; writing: S.P.C.; review and editing: R.R., C.R.d.A. and P.P.; visualization: S.P.C.; supervision: C.R.d.A. and P.P. All authors have read and agreed to the published version of the manuscript.

**Funding:** This research received no external funding.

**Institutional Review Board Statement:** Not applicable.

**Informed Consent Statement:** Not applicable.

**Data Availability Statement:** Data Availability Statements in section "Scopus: https://www.scopus.com/search/" and "Web of Science: https://www.webofscience.com/wos/woscc/basic-search" (accessed on 5 September 2022).

**Acknowledgments:** We are grateful to CEFAGE (Center for Advanced Studies in Management and Economics, University of Évora) and to CinTurs (Research Center for Tourism, Sustainability and Well-Being), University of the Algarve.

**Conflicts of Interest:** The authors declare no conflict of interest.

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
