# Peer review of "Sustainable Drive Tourism Routes: A Systematic Literature Review"

_socsci, doi:10.3390/socsci11110510_

Round 1

Reviewer 1 Report

Dear authors

I make some considerations about your article:

The topic is interesting, still with many possibilities for study.

Summary: well organized; meets the requirements that are needed for this topic.

Introduction: It contextualizes the study theme, presents data to support the study's justification. The purpose of the study is presented. At the end, it presents the organization of the article.

Materials and Methods: Although it is a review article, it presents this topic in detail, which greatly enriches the article. A PRISMA methodology and VOSviewer software are used.

Results: The presentation of results is carried out in a clear and objective way

Discussion: It is divided into subtopics, according to the three components of sustainability: economic, environmental, and social, which was good for organizing the contents, providing a good foundation and updated references.

Conclusions: Objective, it could better punctuate the studies that could be originated from this study, outlining guidelines for future studies.

Yours sincerely.

Author Response

Revisor 1 comments

Thank you for your comments which the authors have agreed. Thus, I add in conclusions, possible studies that could give us some knowledge and insight to improve EN2 route in Portugal.

The English language was revised.

Reviewer 2 Report

The article takes up a very interesting topic. It is a literature study, which always gives great generalizations. However, if one were to try to analyze a particular tourist region, not all generalizations would fit.
In my opinion, it would be worthwhile to ask the analysis in relation to at least specific continents. What is different is drive tourism in Asia than in North America and Australia.

Although the authors write that they analyze the impact of drive tourism on sustainability, they don't do exactly that. Of course, this is clear from the analyzed literature, but they could have commented that there are still too few works on the subject.

One thing I liked was the authors' social perspective on the benefits of drive tourism development.

Author Response

Response to Revisor 2 Comments

The authors pointed out, after the revision, that there are still few works on sustainability and Drive tourism routes.

The authors add in conclusions, possible studies that could give us some knowledge and insight to improve EN2 route in Portugal.

The English language was revised.

Reviewer 3 Report

Drive tourism is an important sector of various tourism activities in shaping sustainable development. I think this paper is valuable for it describe the progress of drive tourism. However, this manuscript should be improved before being published.

First, there are too much space put into discussing sustainable tourism in the beginning of introduction section (line 25-80). I suggest to shorten this part to one paragraph and elaborate more on the significance of drive tourism in sustainable development.

 Second, why do the authors emphasize “route” in the title and in the main body? In searching literature, route is not that important as key word.

 Third, the literature searching process is sound and clear. However, the method and process of data analysis is not fully revealed, especially about the results of discussion section. Is there any guide on data analysis in the PRISMA methodology?

 Fourth, I think the authors should explain more about figure 6-8. The explanative narratives are few in current writing.

Generally, I suggest a minor revision to this submission.

Author Response

Response to Revisor 3

First, there are too much space put into discussing sustainable tourism in the beginning of introduction section (line 25-80). I suggest to shorten this part to one paragraph and elaborate more on the significance of drive tourism in sustainable development.

At this point we made some changes and cut some paragraphs

Second, why do the authors emphasize “route” in the title and in the main body? In searching literature, route is not that important as key word.

The authors aim to study the DT associated with the development of “Drive tourism routes” although there are still few studies in this subject. Therefore, they wish to contribute to this new perspective

Third, the literature searching process is sound and clear. However, the method and process of data analysis is not fully revealed, especially about the results of discussion section. Is there any guide on data analysis in the PRISMA methodology?

The authors used the PRISMA tool or guidelines that uses a set of methods to systematically search papers and literature for review-based studies. This methodology is based on the formulated inclusion and exclusion criteria by the authors for each study. The selection process is explained by the PRISMA 2020 flow diagram.

Fourth, I think the authors should explain more about figure 6-8. The explanative narratives are few in current writing.

The authors made some changes to explain it with more details
